# Association between Daytime Sleepiness, Fatigue and Autonomic Responses during Head-Up Tilt Test in Multiple Sclerosis Patients

**DOI:** 10.3390/brainsci13091342

**Published:** 2023-09-19

**Authors:** Monika Zawadka-Kunikowska, Łukasz Rzepiński, Mirosława Cieślicka, Jacek J. Klawe, Małgorzata Tafil-Klawe

**Affiliations:** 1Department of Human Physiology, Nicolaus Copernicus University, Ludwik Rydygier Collegium Medicum in Bydgoszcz, Karłowicza 24, 85-092 Bydgoszcz, Poland; m.cieslicka@cm.umk.pl (M.C.); malg@cm.umk.pl (M.T.-K.); 2Sanitas–Neurology Outpatient Clinic, Dworcowa 110, 85-010 Bydgoszcz, Poland; luk.rzepinski@gmail.com; 3Department of Neurology, 10th Military Research Hospital and Polyclinic, 85-681 Bydgoszcz, Poland; 4Department of Hygiene, Epidemiology, Ergonomy and Postgraduate Education, Nicolaus Copernicus University, Ludwik Rydygier Collegium Medicum in Bydgoszcz, M. Sklodowskiej-Curie 9, 85-094 Bydgoszcz, Poland; jklawe@cm.umk.pl

**Keywords:** multiple sclerosis, fatigue, excessive daily sleepiness, sleepiness, autonomic dysfunction, blood pressure

## Abstract

We aimed to assess dynamic changes in hemodynamic and autonomic function in response to the head-up tilt test (HUTT) in patients with multiple sclerosis (MS) compared to healthy controls (HCs) and evaluate its relationship with the patients’ reported daytime sleepiness and fatigue symptoms. A total of 58 MS patients and 30 HCs were included in the analysis. Fatigue and sleepiness were evaluated using the Chalder Fatigue Scale (CFQ) and the Epworth Sleepiness Scale (ESS), respectively. Hemodynamic response, baroreflex sensitivity, heart rate variability, and systolic and diastolic blood pressure (BP) variability (SBPV, DBPV) parameters were calculated at rest, and in response to the HUTT. The MS patients displayed attenuated BP responses coupled with a more pronounced decrease in cardiac index as well as a reduced increase in the low frequency (LFnu) of DBPV (*p* = 0.021) and the sympathovagal ratio (*p* = 0.031) in the latter-phase orthostatic challenge compared to HCs. In MS patients, the ESS score showed no correlation with CFQ or clinical disease outcomes, but exhibited a moderate correlation with LFnu of BPV_rest_. Fatigue and disease variants predicted blood pressure response to HUTT. These findings underscore the importance of subjective daytime sleepiness and fatigue symptoms and their role in blood pressure regulation in MS patients.

## 1. Introduction

Multiple sclerosis (MS) is characterized by focal as well as diffuse damage to the central nervous system (CNS), causing alterations in the autonomic nervous system (ANS) on the level of cardiac and vascular control [1,2,3]. During the course of the disease, about 50% of patients, including those with normal autonomic reflexes, report complaints of orthostatic intolerance (OI), accompanied by non-specific symptoms such as dizziness, fatigue, and vision changes [4,5,6]. In MS, OI syndromes, including orthostatic hypotension (OH), and postural tachycardia syndrome (POTS) represent distinct manifestations of cardiovascular autonomic dysfunction (CAD) that manifest as abnormal blood pressure (BP) and heart rate (HR) responses to changes in body posture or prolonged standing [7,8].

Recent observations have shown that the risk of developing CAD increases with age, the progressive course of the disease, longer disease duration, greater disability, and fatigue symptoms [6,9,10,11,12]. Orthostatic dysregulation is not typically the initial manifestation of MS; however, it is associated with an increased risk of falls, cardiovascular events, cerebral hypoperfusion, and cognitive impairment [1].

Recently, there has been a growing interest in the non-motor symptoms of MS, such as fatigue, excessive daytime sleepiness (EDS), and depression, due to their overlapping features and serious impact on patients’ daily functioning and quality of life [13,14]. Clinically distinguishing fatigue from tiredness and excessive daytime sleepiness (EDS) can be challenging, especially considering that somnolent MS patients additionally report a high severity of fatigue symptoms [13,15]. Sleepiness, which appears to be an overlooked symptom in MS, is characterized by difficulties staying awake and alert throughout the day. It is less common compared to fatigue, with a prevalence ranging between 19 and 45% [13,16,17]. Importantly, EDS is associated with an increased risk of cardiovascular disease, stroke, and mortality, and correlated with fatigue symptoms and vision dysfunction in MS [18,19,20]. While EDS may originate directly from a wide range of sleep or circadian rhythm disorders, recent observations have revealed weak correlations between objective daytime sleepiness and self-reported EDS questionnaires, suggesting multifactorial pathogenesis. [15,21,22,23]. Additionally, some studies have described the relationship between excessive daytime sleepiness (EDS) and autonomic parameter activity in different cardiorespiratory (obstructive sleep apnea, hypertension) [24,25] and neurological (spinal cord injury, Parkinson’s disease) [26,27] patient cohorts. Nevertheless, the physiological mechanism behind this association remains unclear.

The head-up tilt test (HUTT) is widely used in clinical practice with various protocols for diagnosing different orthostatic intolerance (OI) symptoms, as well as characterizing changes in cardiovascular sympathetic and parasympathetic autonomic functions [7,28]. Previous studies have shown that the mechanism of OI in MS patients may be caused by lesions in the ANS structures and impaired hemodynamic response, which depends on the peripheral responsiveness of mechanoreceptors, including baroreceptors [29]. The majority of previous studies have focused only on cardiac autonomic parameters, such as HR and/or BP fluctuations [4,6,9,12]. The use of additional cardiovascular parameters based on impedance cardiography (ICG) provides a more comprehensive assessment in analyzing the results of the HUTT. Progressive monitoring, through various time points, may indicate abnormal changes in cardiac function or peripheral resistance and highlight compensatory adjustments in other functions [30]. Based on these considerations, a greater understanding of the interplay of the sequence of beat-to-beat hemodynamic and autonomic responses during orthostatic challenges and their relationship with fatigue and daily sleepiness symptoms is crucial for enhancing cardiovascular outcomes. Thus, we aimed to assess dynamic changes in hemodynamic and autonomic function in response to the head-up tilt test in patients with MS, compare it to HCs, and evaluate its relationship with patients’ reported daily sleepiness and fatigue symptoms.

## 2. Materials and Methods

This study population consisted of 58 MS patients and 30 age-matched healthy controls (HCs) from 2017 to 2023. All the research procedures were approved by the review of the Bioethical Committee of Collegium Medicum in Bydgoszcz, Nicolaus Copernicus University in Torun (reference number KB 747/2017). All the subjects provided written informed consent to participate in the study. The diagnosis of MS was established by neurologists based on the 2017 McDonald and Polman et al. criteria [31,32]. From the 58 subjects with MS, the relapsing–remitting variant (RRMS) and progressive variants (primary progressive, PPMS; and secondary progressive, SPMS) were included. Healthy subjects serving as controls (Table 1) were recruited from the local community of Bydgoszcz, Poland. For each subject, a complete medical and medication history was taken and a physical examination was performed, with emphasis on the symptoms and signs of MS. Mild, moderate, and severe disability were defined as a Kurtzke Expanded Disability Status Scale (EDSS) score of mild (EDSS ≤ 3.5), moderate (4.0 ≤ EDSS ≤ 5.5), and severe categories (EDSS ≥ 6.0), respectively [33].

Inclusion criteria for MS and HC subjects were as follows: (a) definite MS and EDSS disability score ≤ 8, no clinical markers of disease exacerbation during the 90 days preceding the study (this was defined as the occurrence of new or worsening of previously reported MS-related symptoms lasting for more than 24 h and resulting in an increase in the EDSS score of at least 1.0 point if the baseline EDSS was ≤5.5, or an increase in the EDSS by at least 0.5 points if the baseline EDSS was above 5.5 [34]); (b) no history of cardiovascular, metabolic disease (diabetes mellitus), malignancy, or acute or chronic infection; and (c) no current treatment with beta-blockers or antiarrhythmics. Controls who manifested central or peripheral nervous system lesions and any other disease known to affect the autonomic nervous system were excluded.

### 2.1. Protocol

All subjects were asked to avoid intense physical activity, smoking, caffeine, and alcohol for 12 h before the test, but they were allowed to consume beverages (water) and take their medications. The investigation was performed at our autonomic laboratory between 8 and 12 a.m. in a quiet, darkened room with a stable temperature of 22 ± 1 °C and air humidity [35]. All parameters were recorded after this stabilization period in the supine position at rest for 10 min, followed by a 7 min and 20 s head-up tilt test (HUTT) at 70°. During the HUTT, the parameters were recorded at different time intervals: 1 min 20 s (phase_1_), 3 min 20 s (phase_2_), 5 min 20 s (phase_3_), and 7 min 20 s (phase_4_). We calculated the changes (Δ) in the analyzed parameters at different time intervals from the supine to the tilting position (Δphase-supine).

A Task Force Monitor (CNSystems, Graz, Austria) was used for beat-to-beat cardiovascular autonomic recordings. Continuous systolic (sBP), diastolic (dBP), and mean (mBP) blood pressure were measured on the right hand (using the second and third fingers) using a vascular unloading technique, which was automatically compared to the oscillometric blood pressure measured at the left brachial artery. The heart rate (HR) was calculated continuously from an electrocardiogram (ECG) [36]. Impedance cardiography was used to evaluate the cardiac index (CI = CO/body surface). The total peripheral resistance index (TPRI) was calculated according to Ohm’s law: total peripheral resistance index = mean BP/cardiac index [37]. The following HRV and BPV parameters were calculated through power spectral analysis: total power spectral density (PSD-RRI) (<0.40 Hz), low-frequency power (LF) in the range of 0.04–0.15 Hz, high-frequency power (HF) in the range of 0.17–0.40 Hz, LF and HF in normalized units (LFnu, HFnu), and the ratio between LF and HF bands (LF/HF ratio) for both HRV and BPV, which represented the sympathetic–parasympathetic balance [35] from beat-to-beat BP and HR monitoring. Baroreceptor sensitivity (BRS) was calculated using the spontaneous sequence method as the slope of the linear regression between beat-to-beat sBP values (mmHg) [38]. The TFM software utilized the adaptive autoregressive model (AAR) proposed by Bianchi et al. [39] and a recursive least-squares algorithm to assess the power spectral analysis for both HRV and blood BPV. The HF spectral power of HRV is regarded as an indicator of parasympathetic nervous system activity, notably influenced by breathing. Conversely, LF fluctuations often reflect the combined interplay of both sympathetic and parasympathetic modulations. The low frequency (LF) of blood pressure variability (BPV) is influenced by sympathetic modulation of vascular tone and myogenic vascular function [36,40].

In our study, the data extracted for HRV and BPV were exported from the TFM program to Microsoft Excel for further analysis, followed by the import of all the data into Statistica 13. The AAR model could lead to outliers; therefore, beat-to-beat data were filtered using Grubbs’s test to remove artifacts. This method of filtering is well documented and has a strong mathematical background [41].

### 2.2. Fatigue and Excessive Daytime Sleepiness Questionnaires

In all subjects, we evaluated fatigue and EDS symptoms using the following questionnaires.

The Chalder Fatigue Scale (CFQ) is an 11-item self-administered scale (range 0–33 score) for measuring the extent and severity of fatigue. Each item is scored 0–3: less than usual (0), no more than usual (1), more than usual (2), and much more than usual (3). CFQ scores ≥ 9 represent high levels of fatigue [42]. Daytime sleepiness was evaluated using the Epworth Sleepiness Scale (ESS), with scores of >10 indicating excessive daytime sleepiness. Each question consisted of a 4-point Likert scale ranging from 0 to 3, with total scores ranging from 0 to 24 [43].

### 2.3. Statistical Analysis

Statistical analyses were performed using Statistica SPSS version 13.3 software. Data were expressed as the mean (standard error, SE) or the median (interquartile range, IQR). Categorical variables were presented as absolute and relative frequency (%). Differences in the distribution of categorical variables were determined using the chi-square test or Fisher’s exact test, while the differences in continuous variables were determined using Student’s *t*-test or the Mann–Whitney U test, depending on the normality of the data distributions. Changes associated with the HUTT were analyzed using the nonparametric Friedman for repeated measures test, followed by Dunn’s multiple comparison test. The strength and significance of the correlation between the selected variables were calculated using the nonparametric Spearman’s test. In the context of multiple regression analysis, we explored the relationship between autonomic indices at rest and the subsequent predictors, selected based on previous analyses: sleepiness measured by ESS, fatigue assessed by CFQ, age, disease variant (RRMS vs. PMS), and EDSS. A *p*-value of less than 0.05 was considered statistically significant.

## 3. Results

Table 1 summarizes the demographic characteristics of the 88 subjects included. A total of 58 MS patients were included (mean age of 47.45 years ± 11.90; mean duration since diagnosis, 9.90 years; progressive variants of MS (43.38%)). Thirty-four (58.62%) patients presented with a relapsing–remitting course of MS (RRMS), 8 (13.79%) with a primary progressive (PPMS) and 16 (27.59%) with a secondary progressive (SPMS) course. Most MS patients were in the mild disability (50.00%) and moderate disability (34.48%) categories. The patients in the RRMS group compared to the PMS group were significantly younger (40.79 ± 10.78 vs. 52.08 ± 8.09, *p* < 0.001), with a shorter disease duration (8.29 ± 6.62 vs. 12.17 ± 6.88, *p* < 0.035) and a lower EDSS score (2.37 ± 1.38 vs. 5.13 ± 0.99, *p* < 0.001), respectively.

All subjects had sinus rhythm at the baseline position. The baseline values of cardiac parameters, BRS, BPV, and HRV were not significantly different between the MS and HC groups, at *p* > 0.05 (Table 2). Both groups demonstrated preserved cardiovascular autonomic function during HUTT testing. However, five patients (8.62%) and one of the HCs (3.33%) exhibited test results consistent with POTS, characterized by a sustained heart rate increase of 30 beats/min upon standing, without signs of OH. Furthermore, OH [7] was identified in four (6.89%) MS patients. The HR, sBP, dBP, mBP, and TPRI values increased significantly during all phases_1–4_ of the 7 min tilt compared to the supine position in both groups: *p* < 0.05 (Figure 1). The MS group, compared to the HCs, showed a lower post-tilt increase in blood pressure at the 5 min 20 s of the HUTT: ΔdBP in phase_1_ (*p* = 0.021) and phase_3_ (*p* = 0.037); ΔsBP (*p* = 0.04), ΔdBP (*p* = 0.001), and ΔmBP (*p* = 0.003) in phase_2_. In addition, the MS group was characterized by a greater post-tilt decrease in the cardiac index (CI) during the entire period of tilt: ΔCI phase_1_ (*p* = 0.043), ΔCI phase_2_ (*p* = 0.03), ΔCI phase_3_ (*p* = 0.021), and ΔCI phase_4_ (*p* = 0.035) than the HC group (Table 3).

During the HUTT, the MS group, in contrast to the HCs, initially showed a non-significant fall in LFnu-RRI and sympathovagal balance (LF/HF, LF/HF-RRI) and an increase in HFnu-RRI until 1 min and 20 s of the HUTT (phase_1_), followed by a significant increase in the latter phases of the tilt (Phases_2–4_) compared to the baseline. Both groups showed a significant fall in LFnu-dBP until 1 min and 20 s (phase_1_) and a later increase in LFnu-dBP and HFnu-dBP at the end of the HUTT (phases_3–4_) compared to the baseline. Similarly, significant increases in the LFnu-sBP and HFnu-sBP values were also observed in all phases of the HUTT compared to the supine position. The pattern of PSD decrease was similar in both groups. The MS group, compared to the HCs, showed a lower post-tilt increase in the ΔLF/HF (*p* = 0.031) and the ΔLFnu of dBPV (*p* = 0.035) at the end of the HUTT (phase_3_, phase_4_, respectively) than the HC group (Figure 1). Sleepiness in patients with MS was not different from that in HCs: *p* > 0.05 (Figure 1 and Figure 2; Table 2 and Table 3).

### 3.1. Patient-Reported Sleep and Fatigue Symptoms

Overall, the means of the ESS scores were 5.62 ± 4.25 and 5.60 ± 3.73, which was in the normal range in MS patients and HCs, respectively. A total of 24.14% of MS patients and 16.67% of HCs had an EDS (ESS > 10). The prevalence and severity of a pathological ESS score was similar between MS subgroups (*p* = 0.291) and between patients and HCs: *p* > 0.05 (Table 1).

Among the MS patients, 72.41% (42/58) experienced only fatigue and 24.14% (14/58) had fatigue and sleepiness symptoms. The MS patients had higher CFQ scores compared to HCs (*p* < 0.001). There were no differences in the frequency of fatigue symptoms between the MS subgroups, as well as between the patients and HCs (Table 1). In total, 95% of MS patients and 83% of HCs were categorized as fatigued.

### 3.2. Relationships between Daily Sleepiness, Fatigue Symptoms, Clinical, Demographic, and Cardiovascular Autonomic Parameters at Rest, and in Response to Orthostatic Challenges

In the MS patients, fatigue symptoms, expressed as the CFQ scale, were positively correlated with EDSS (R = 0.27; *p* = 0.04), as well as negatively correlated with lower post-tilt increase of sympathovagal balance, i.e., ΔLF/HF-RRI in phase_2_ (R = −0.31; *p* = 0.016), ΔLF/HF in phase_2_ (R = −0.36; *p* = 0.005), and ΔLF/HF in phase_4_ (R = −0.26; *p* = 0.045) (Appendix A). Overall, there was no association between fatigue and disease duration and age of the subjects. Excessive daytime sleepiness as determined by ESS was negatively correlated with age (R = −0.33, *p* = 0.011), sympathovagal balance (R = −0.28, *p* = 0.033), and variability of LFnu-sBP (R = −0.42, *p* = 0.001), LFnu-dBP (R = −0.32, *p* = 0.013) at rest, and ΔPSD-RRI (R = −0.26, *p* = 0.047) in phase_2_ and positively with ΔdBP in phase_3_ (R = 0.28, *p* = 0.034). No significant correlations were found between ESS and fatigue, sex, EDSS, or disease duration, *p* > 0.05 (Appendix A). In the multivariable regression model, the ESS score adjusted for age, CFQ, EDSS, and disease variant (RRMS/PMS) predicted LFnu-sBP (b = −0.41; *p* = 0.003, R^2^ = 0.17), explaining 22% of the variance. Similarly, age (b = −0.40; *p* = 0.01) and CFQ (b = −0.39; *p* = 0.003) were identified as independent predictors of ΔLF/HF in phase_4_, and explaining 25% of the variance. The CFQ score (b = 0.38; *p* = 0.002, b = 0.31, *p* = 0.019) and disease variant (RRMS/PMS, b = −0.44; *p* = 0.016, b = −0.49; *p* = 0.014) predicted ΔdBP and ΔsBP in phase_4_ (R^2^ = 0.37, R^2^ = 0.23), respectively. 

## 4. Discussion

Our major findings are as follows: (1) the MS patients displayed a lower blood pressure response coupled with a more pronounced decrease in cardiac index, a decreased sympathovagal ratio, and an altered diastolic blood pressure variability in response to the orthostatic challenge compared to HCs; (2) in the patient group, the ESS score exhibited no significant correlation with CFQ or clinical disease outcomes, but exhibited a mild to moderate correlation with LFnu of BPV_rest_; (3) fatigue and disease variant (relapsing/progressive) predicted blood pressure response to HUTT.

The normal compensatory response to passive upright tilt includes an increase in mean arterial pressure and decreases in stroke volume and cardiac output. Additionally, the rapid inactivation (unloading) of the inhibitory arterial baroreflexes, driven by enhanced sympathetic activity, increases the heart rate and total peripheral resistance, a mechanism identified in our study [44,45]. The frequency of POTS during HUTT testing was lower (9%) than reported by Adamec et al. [46] (19%) in a population of 293 Croatian MS patients. The variation could be ascribed to the potential age discrepancy (35.68 years vs. 47.45 years) as well as potential disparities in the MS phenotype. Considering the underlying mechanism, POTS is likely to arise from both increased sympathetic activity and an autoimmune process [44,45]. Notably, four MS patients fulfilled the criteria for classic OH diagnosis, indicating a potential interplay between the compromised responsiveness of mechanoreceptors and chemoreceptors, along with brainstem dysfunction [47,48].

A lower increase in blood pressure during tilt in MS patients may be related to a greater reduction in cardiac index due to limited preload by venous pooling in the lower limbs and pelvis [44]. Thus, a compensatory increase in total peripheral resistance may suggest that the efferent sympathetic pathways to the peripheral vessels were intact. In this context, a lower increase in both sympathovagal balance and sympathetic vasomotor activity, as shown by LFnu-dBP, in the MS subjects during orthostasis is also a possible explanation for the lower blood pressure responses. Interestingly, the MS and HC groups showed opposite changes in the balance between sympathetic and parasympathetic outflows shortly after the initiation of the orthostatic challenge (1 min and 20 s) during the HUTT; that is, the MS patients demonstrated diminished activation of LFnu of HRV and decreased sympathovagal balance during the early phase of the HUTT. However, these changes did not differ significantly from the supine position in both groups. As assessed by its chronotropic effect, the baroreceptor reflex response to the HUTT was preserved among the MS and HC groups [49,50,51]. The physiological mechanisms underpinning CAD among individuals with MS are not well understood. Plausible pathways involve the CNS potentially originating from brainstem lesions and associated neurodegenerative processes, as well as peripheral contributions via the interplay between inflammation-induced modulation of the sympathetic nervous system and the peripheral discharge of catecholamines, particularly through beta 2-adrenergic receptor activation [51].

Recent studies have revealed a significant correlation between fatigue and CAD [51,52]. As previously reported, the present study demonstrated that MS patients were more fatigued than HCs, and both fatigue and age were inversely correlated with LF/HF response to the HUTT. Accordingly, Keselbrener et al. compared 10 fatigued MS patients with 10 HCs and found a significant increase in LF and LF/HF of HRV upon standing in control subjects only. They concluded that there was a possible impairment of the sympathovagal balance response to standing in patients with MS who experienced fatigue [12]. Therefore, the authors assumed that age-related reduction in vagal activity occurred earlier in patients with MS who were fatigued. In line with these findings, Flachenecker et al. documented a connection between fatigue and a hypoadrenergic orthostatic response in their investigation involving 60 MS patients. This correlation could potentially be linked to sympathetic vasomotor dysfunction, which might also explain the observed lower LFnu-dBP values in our study [6]. Another study (46 patients with secondary MS, 46 with CIS, 44 HCs) confirmed that patients with the secondary variant of MS may exhibit SBPV alterations compared to individuals with CIS and normal controls. They observed lower LF [ms^2^] SBPV and LF/HF during the HUTT in secondary MS patients compared to both the CIS and HC groups, which may partly relate to our study [53].

We also observed that the disease variant and fatigue were independent predictors of the systolic–diastolic BP response to the orthostatic challenge. These findings are consistent with previous studies that link the progressive MS variant with reduced sympathetic reactivity during head-up tilt [9,54]. As previously reported, we also noted that the progressive MS variant has been associated with lower sympathetic reactivity during head-up tilt [10,50]. In contrast, in the study by Gervasoni et al. [55], sympathovagal parameters during the orthostatic challenge were similar in the PPMS and RRMS groups, and in the PPMS and HC groups. The progression of autonomic nervous system (ANS) dysregulation in MS entails a shift from mildly diminished sympathetic activity during the initial stages to a chronic phase characterized by increased basal sympathetic tone and autonomic imbalance. This state is frequently coupled with diminished sympathetic responsiveness to orthostatic challenges and a deficiency in parasympathetic function, as noted in progressive MS variants [52].

The prevalence of EDS in the general population varies from 4% to 20.6% [56]. In line with previous studies, subjective EDS (ESS > 10) occurred less frequently than fatigue, appearing in 19% of patients with MS; however, its frequency and severity were similar [57] to those observed in HCs. Prior research indicates a potential association between the presence of EDS among individuals with MS and sleep disorders, including sleep-related breathing disorders, restless leg syndrome, and chronic insomnia. Notably, these sleep disturbances have been postulated as potential indicators of fatigue [21,58], or exacerbate sleepiness in fatigued patients [15,17]. A recent review of 48 studies demonstrated a moderate association between sleepiness, as indicated by ESS scores, and various fatigue rating scales [15]. However, this finding is not consistent with our results. Our analysis revealed no associations between ESS and fatigue severity, and clinical disease outcomes, suggesting that EDS and fatigue are distinct conditions [15,17,23,28]. A few studies have suggested that the degree of sleepiness can exhibit significant variation, presenting itself as either elevated or diminished, irrespective of fatigue levels [16]. We only recruited patients meeting the criteria for ANS testing, as well as using different measures of fatigue, which serve as a partial explanation for the lack of relationship between fatigue and ESS.

Our preliminary results demonstrated a significant relationship between the ESS score, age, and BPV parameters at rest. Although the model’s R-squared value was relatively low (R^2^ = 0.18 to 0.19), the higher ESS score exhibited a mild to moderate correlation, with lower values in the LF component of both systolic and diastolic BPV at rest, reflecting measures of sympathetic vasomotor activity. The physiological mechanism underlying this association remains unclear; however, the potential involvement of altered sympathetic activity, as well as baroreflex modulation, has been suggested [24]. Similarly, in patients with high spinal cord injury lesions (T3 and above), higher daily sleepiness accompanied by reduced sympathetic activity (LF), as measured by HRV, was also observed [26]. Moreover, a few studies have described lower ESS scores in hypertensive patients with OSA, suggesting a possible role of increased sympathetic reactivity. The absence of sleepiness may result from a significant increase in central neural arousal networks that promote daytime wakefulness, driven by adrenergic inputs [25,59]. Nonetheless, findings derived from subjective EDS should be approached cautiously, given the potential for patients to overestimate them compared to objective assessments [15,59].

## 5. Limitations

Some limitations need to be noted. First, we did not analyze the PMS and RRMS separately regarding cardiovascular autonomic responses. Second, we did not compare the patients’ reported orthostatic symptoms with the objective results obtained from the HUTT. Third, we cannot completely exclude the possibility of selection bias due to sex differences between the groups. The association of autonomic function and both objective and self-reported measures of sleepiness should be examined in future studies. Furthermore, we did not account for depression, medication side effects, deconditioning, and pain, which are recognized factors contributing to changes in both autonomic function and fatigue in individuals with MS.

## 6. Conclusions

The MS patients exhibited a lower blood pressure response coupled with a more pronounced decrease in the cardiac index, a decreased sympathovagal ratio, and altered blood pressure variability in response to orthostatic challenges compared to HCs. Within the MS patient group, subjective daytime sleepiness exhibited no association with fatigue symptoms, but it exhibited a moderate correlation with diastolic blood pressure variability at rest. Fatigue and disease variant are potential factors influencing the blood pressure response to HUTT. These findings underscore the importance of subjective daytime sleepiness and fatigue symptoms and their role in blood pressure regulation in MS patients.

## Figures and Tables

**Figure 1 brainsci-13-01342-f001:**
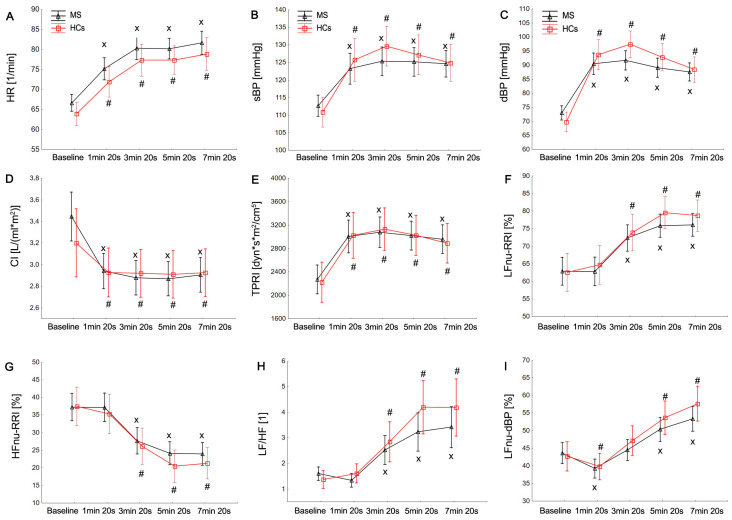
Mean values (±SE) of HR, heart rate (**A**); sBP, systolic blood pressure (**B**); dBP, diastolic blood pressure (**C**); CI, cardiac index (**D**); TPRI, total peripheral resistance index (**E**); LFnu-RRI, low-frequency R-R interval in normalized units (**F**); HFnu-RRI, high-frequency R-R interval in normalized units (**G**) LF/HF, the ratio between low and high band for heart rate and blood pressure variability (**H**); LFnu-dBP, low frequency of diastolic blood pressure variability in normalized units (**I**); and HFnu-dBP, compared to healthy controls (HC). Statistically significant differences between the rest and tilt phases are indicated with # and x in HCs and MS patients, respectively.

**Figure 2 brainsci-13-01342-f002:**
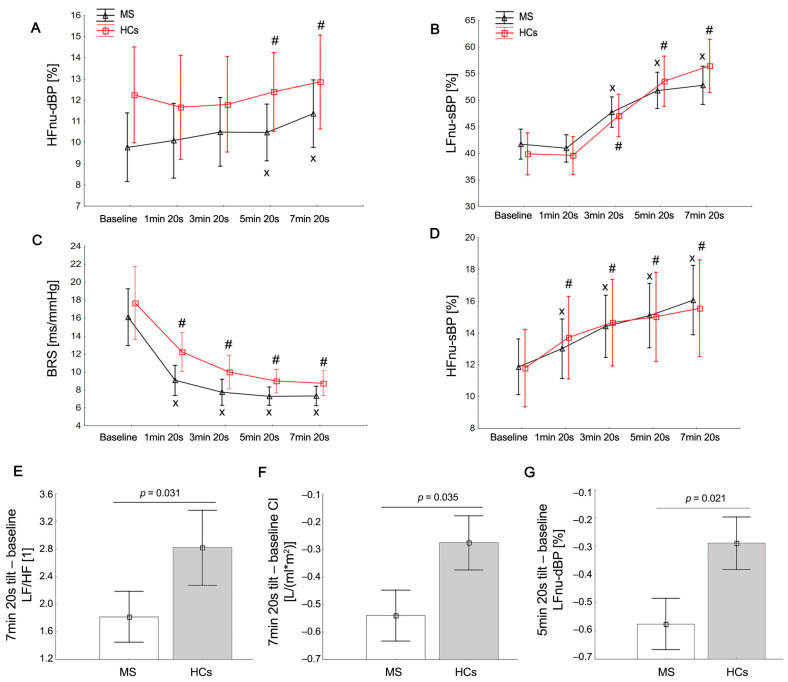
Mean values (±SE) of HFnu-dBP, high frequency of diastolic blood pressure variability in normalized units (**A**); LFnu-sBP, low frequency of systolic blood pressure variability in normalized units (**B**); HFnu-sBP, high frequency of systolic blood pressure variability in normalized units (**C**); BRS, baroreflex sensitivity; (**D**), post-tilt change in LF/HF, the ratio between low and high band for heart rate and blood pressure variability (**E**); post-tilt change in CI, cardiac index (**F**); and post-tilt change in LFnu-dBP, low frequency of diastolic blood pressure variability in normalized units (**G**), compared to healthy controls (HC). Statistically significant differences between the rest and tilt phases are indicated with # and x in HCs and MS patients, respectively.

**Table 1 brainsci-13-01342-t001:** Demographic and clinical data of the study participants.

	MS	HCs	*p*-Value
Number of subjects	58	30	
Sex, female n (%)	47 (81.03)	16 (53.33)	0.006
Age, mean (years)	47.45 ± 11.90	40.53 ± 14.20	0.079
Age at onset, mean (years)	35.55 ± 10.37		
Disease duration MS (years), mean (range)	9.90 ± 6.94 (0.5–28)		
EDDS	3.51 ± 1.84 (0.5–7.00)		
Mild (≤3.5), n (%)	29 (50.00)		
Moderate (4–5.5) n (%)	20 (34.48)		
Severe (>6.0) n (%)	9 (15.52)		
MS variant, n (%)			
RRMS	34 (58.62)		
SPMS	16 (27.59)		
PPMS	8 (13.79)		
Localization of the First Demyelinating Lesions, n (%)			
Supratentorial + optic nerves	40 (68.97)		
Spinal cord	14 (24.14		
Cerebellum	3 (5.17)		
Brainstem	1 (1.72		
Type of DMD n (%)			
Interferon-beta	8 (13.79)		
Glatiramer acetate	4 (6.89)		
Dimethyl fumarate	2 (3.44)		
Natalizumab	1 (1.72)		
CFQ, mean (score)	17.81 ± 6.08	12.63 ± 5.91	<0.001
RRMS	17.03 ± 5.98		
PMS	18.92 ± 6.18		
ESS, mean (score)	5.62 ± 4.25	5.60 ± 3.73	0.999
RRMS	5.68 ± 4.37		
PMS	5.54 ± 14.7		
Daily sleepiness, n (%)	11 (18.97)	3 (10)	0.275
RRMS	8 (23.83)		
PMS	3 (12.50)		
Fatigue symptoms, n (%)	55 (94.83)	25 (83.33)	0.075
RRMS	32 (55.17)		
PMS	23 (39.66)		

**Table 2 brainsci-13-01342-t002:** Mean (±SE) of resting and during-tilt-test cardiac autonomic measures for patients with MS and healthy controls (HC).

Variables	Group	Baseline	1.20 Phase I	3.20 min Phase II	5.20 min Phase III	7.20 Phase 4
HR [1/min]	MS	66.61 ± 0.93	75.10 ± 1.28	80.29 ± 1.38	80.14 ± 1.32	81.59 ± 1.37
	Control	63.87 ± 1.77	71.90 ± 2.17	77.27 ± 2.14	77.28 ± 1.71	78.82 ± 2.25
sBP [mmHg]	MS	112.67 ± 1.53	123.22 ± 2.19	125.34 ± 2.20	125.16 ± 2.14	124.64 ± 1.97
	Control	110.84 ± 2.10	125.74 ± 3.13	129.64 ± 2.39	127.17 ± 2.64	124.91 ± 2.42
dBP [mmHg]	MS	73.02 ± 1.17	90.52 ± 1.90	91.68 ± 1.76	89.10 ± 1.71	87.60 ± 1.57
	Control	69.76 ± 1.99	93.75 ± 2.72	97.44 ± 2.21	92.84 ± 2.42	88.47 ± 2.40
mBP [mmHg]	MS	89.69 ± 1.27	104.28 ± 1.93	105.84 ± 1.85	104.25 ± 1.79	103.06 ± 1.66
	Control	86.74 ± 1.97	106.70 ± 2.72	110.65 ± 2.08	106.79 ± 2.35	103.13 ± 2.28
CI [L/(min·m^2^)]	MS	3.45 ± 0.13	2.94 ± 0.09	2.88 ± 0.08	2.87 ± 0.08	2.91 ± 0.09
	Control	3.20 ± 0.12	2.93 ± 0.10	2.92 ± 0.10	2.91 ± 0.10	2.93 ± 0.10
TPRI [dyn·s·m^2^/cm^5^]	MS	2272.01 ± 138.87	3006.57 ± 146.91	3078.56 ± 132.68	3021.80 ± 123.70	2959.50 ± 125.04
	Control	2222.31 ± 123.18	3025.78 ± 180.03	3131.44 ± 180.25	3023.11 ± 173.12	2890.67 ± 163.24
LFnu-RRI [%]	MS	62.85 ± 2.06	62.84 ± 1.93	72.35 ± 1.80	75.87 ± 1.55	76.10 ± 1.57
	Control	62.54 ± 2.55	64.70 ± 3.11	73.93 ± 2.86	79.52 ± 2.57	78.71 ± 2.35
HFnu-RRI [%]	MS	37.24 ± 2.05	37.16 ± 1.93	27.65 ± 1.80	24.13 ± 1.55	23.90 ± 1.57
	Control	37.46 ± 2.55	35.30 ± 3.11	26.07 ± 2.86	20.48 ± 2.57	21.29 ± 2.35
PSD-RRI [ms^2^]	MS	1684.21 ± 214.16	1654.46 ± 247.71	1046.81 ± 139.50	702.05 ± 91.00	647.44 ± 112.41
	Control	2379.92 ± 322.34	2184.37 ± 460.47	1452.78 ± 245.27	1272 ± 182.64	998.87 ± 129.32
LF/HF-RRI [1]	MS	2.51 ± 0.26	2.33 ± 0.21	4.28 ± 0.48	4.99 ± 0.50	4.99 ± 0.61
	Control	2.20 ± 0.22	2.86 ± 0.50	4.76 ± 0.73	6.24 ± 0.88	6.07 ± 0.93
LF/HF [1]	MS	1.60 ± 0.15	1.34 ± 0.10	2.52 ± 0.28	3.23 ± 0.33	3.42 ± 0.40
	Control	1.37 ± 0.11	1.61 ± 0.25	2.84 ± 0.41	4.20 ± 0.63	4.19 ± 0.57
LFnu-dBP [%]	MS	43.58 ± 1.62	39.22 ± 1.46	44.47 ± 1.59	50.33 ± 1.83	53.36 ± 1.90
	Control	42.70 ± 1.72	39.81 ± 1.54	47.15 ± 1.83	53.64 ± 2.11	57.62 ± 2.13
HFnu-dBP [%]	MS	9.77 ± 0.62	10.09 ± 0.83	10.50 ± 0.79	10.48 ± 0.58	11.37 ± 0.79
	Control	12.25 ± 1.54	11.67 ± 1.38	11.80 ± 1.21	12.39 ± 1.15	12.86 ± 1.16
PSD-dBP [mmHg^2^]	MS	8.21 ± 0.65	8.00 ± 0.67	6.93 ± 0.61	6.35 ± 0.56	6.26 ± 0.58
	Control	11.45 ± 1.80	12.84 ± 2.22	9.52 ± 1.39	10.06 ± 1.74	9.74 ± 1.68
LFnu-sBP [%]	MS	41.78 ± 1.54	41.59 ± 1.44	47.78 ± 1.48	51.87 ± 1.76	52.80 ± 1.87
	Control	39.92 ± 1.59	39.64 ± 1.61	47.13 ± 1.89	53.61 ± 2.22	56.48 ± 2.30
HFnu-sBP [%]	MS	11.87 ± 0.90	13.45 ± 1.01	14.43 ± 1.06	15.11 ± 1.13	16.08 ± 1.24
	Control	11.79 ± 1.15	13.72 ± 1.15	14.65 ± 1.11	15.02 ± 1.04	15.56 ± 1.05
PSD-sBP [mmHg^2^]	MS	13.55 ± 1.26	11.95 ± 1.10	10.51 ± 0.99	9.92 ± 0.95	9.76 ± 1.01
	Control	18.25 ± 2.84	16.06 ± 2.75	13.95 ± 2.29	12.98 ± 2.06	12.65 ± 1.96
BRS [ms/mmHg]	MS	15.78 ± 1.57	9.16 ± 0.74	8.59 ± 1.07	7.11 ± 0.43	7.61 ± 0.54
	Control	17.69 ± 1.85	12.25 ± 1.29	10.11 ± 1.22	9.11 ± 0.76	8.89 ± 0.81

HR—heart rate; sBP—systolic blood pressure; dBP—diastolic blood pressure; mBP—mean blood pressure; CI—cardiac index; TPRI—total peripheral index; LFnu-RRI—low-frequency R-R interval in normalized units; HFnu-RRI—high-frequency R-R interval in normalized units; PSD-RRI—power spectral density R-R interval; LF/HF—ratio between low and high band for heart rate and blood pressure variability; PSD-sBP—power spectral density of systolic blood pressure variability; LFnu-sBP—low frequency of systolic blood pressure variability in normalized units; HFnu-sBP—high frequency of systolic blood pressure variability in normalized units; PSD-dBP—power spectral density of diastolic blood pressure variability; LFnu-dBP—low frequency of diastolic blood pressure variability in normalized units; HFnu-dBP—high frequency of diastolic blood pressure variability in normalized units; BRS—baroreflex sensitivity.

**Table 3 brainsci-13-01342-t003:** Median [IQR] of post-tilt changes (Δ) in cardiac autonomic measures for patients with MS and healthy controls (HC).

Variables	Group	Δ1.20 Phase_1_-Baseline	3.20 min Phase_2_-Baseline	5.20 min Phase_3_-Baseline	7.20-Phase_4_-Baseline
HR [1/min]	MS	6.70 [2.04; 16.10]	13.01 [5.14; 25.87]	12.45 [7.37; 17.01]	13.17 [9.65; 19.16]
	HCs	7.99 [2.48; 11.08]	14.54 [7.67; 18.78]	14.29 [8.73; 18.05]	14.78 [9.34; 19.11]
sBP [mmHg]	MS	12.70 [−7.56; 21.78]	13.65 [−2.66; 24.89] *	14.83 [7.40; 20.72]	12.09 [6.52; 18.97]
	HCs	15.79 [6.27; 24,28]	19.06 [11.63; 26.59]	15.86 [8.91; 24.61]	12,29 [6.22; 19.39]
dBP [mmHg]	MS	17.50 [2.84; 31.99] *	20.21 [3.84; 31.08] ***	17.93 [11.12; 24.20] *	15.28 [8.40; 21.91]
	HCs	22.41 [16.97; 28.47]	26.92 [21.51; 33.62]	19.44 [15.65; 29.59]	17.51 [14.18; 21.11]
mBP [mmHg]	MS	14.59 [1.66; 25.35]	17.91 [3.22; 27.99] **	16.32 [9.34; 23.10]	13.46 [7.89; 19.88]
	HCs	18.32 [11.65; 26.16]	23.71 [18.72; 31.43]	17.58 [12.18; 25.94]	−2.28 [8.75; 18.09]
CI [L/(min·m^2^)]	MS	−0.52 [−1.34; −0.32] *	−0.57 [−1.48; 0.36] *	−0.70 [−0.95; −0.18] *	−0.62 [−0.95; −0.18] *
	HCs	−0.27 [−0.54; 0.06]	−0.26 [−0.59; 0.07]	−0.26 [−0.53; 0.09]	−1.55 [−0.63; 0.10]
TPRI [dyn·s·m^2^/cm^5^]	MS	737.78 [55.51; 1404.70]	869.46 [7.54; 1737.64]	850.65 [403.19; 1193.66]	689.10 [401.66; 1075.18]
	HCs	874.28 [367.95; 1132.58]	835.74 [525.85; 1171.54]	707.14 [423.84; 1169.55]	596.28 [329.06; 943.47]
LFnu-RRI [%]	MS	0.38 [−6.07; 4.49]	9.15 [−5.76; 26.23]	10.16 [2.47; 23.63]	12.47 [1.90; 24.03]
	HCs	2.68 [−2.26; 8.69]	10.70 [1.26; 18.13]	17.27 [7.30; 25.61]	18.44 [9.17; 24.63]
HFnu-RRI [%]	MS	−0.49 [−10.09; 9.54]	−9.15 [−26.36; 5.76]	−10.16 [−23.63; −2.47]	−12.47 [−24.03, −1.90]
	HCs	−2.68 [−8.69; 2.26]	−10.70 [−18.13; −1.26)	−17.27 [ −25.61; −7.30]	−18.44 [−24.63; −9.17]
PSD-RRI [ms^2^]	MS	−73.79 [−1252.38; 486.99]	−240.83 [−2659.36; 409.77]	−543.85 [−1491.81; −101.39]	−527.51 [−1732.01; −170.71]
	HCs	−79.26 [−0.43; 1.00]	−527.27 [−1262.40; −66.46]	−929.83 [ −1778.30; −167.37]	−1081.45 [−2655.62; −298.57]
LF/HF-RRI [1]	MS	−0.01 [−1.06; 0.81]	0.81 [−0.87; 5.09]	1.90 [0.61; 4.19]	1.16 [0.19; 3.04]
	HCs	0.23 [−0.43; 1.00]	1.71 [−0.10; 3.24]	2.91 [ 0.87; 5.96]	2.82 [0.77; 5.48]
LF/HF [1]	MS	−0.10 [−0.81; 0.21] *	0.44 [−0.64; 3.06]	1.15 [0.28; 2.52]	1.04 [0.18; 2.39] *
	HCs	−0.04 [−0.30; 0.40]	0.91 [0.13; 1.91]	2.02 [0.66; 3.27]	1.65 [1.28; 4.63]
LFnu-dBP [%]	MS	−3.37 [−13.85; 3.69]	0.85 [−8.62; 10.06]	7.27 [1.14; 11.49] *	10.06 [3.73; 15.86]
	HCs	−2.17 [−6.47; 2.38]	3.50 [−0.69; 7.71]	8.46 [ 2.17; 18.98]	14.86 [5.89; 23.73]
HFnu-dBP [%]	MS	−0,18 [−1.83; 2.37]	0.41 [−2.05; 3.65]	0.54 [−0.50; 2.10]	0.88 [−0.52; 2.67]
	HCs	−0.46 [−1.87; 0.89]	0.40 [−1.81; 1.88]	0.98 [−1.92; 3.09]	1.54 [−1.44; 3.20]
PSD-dBP [mmHg^2^]	MS	−0.40 [−2.04; 1.39]	−1.09 [−3.92; 0.48]	−1.70 [−2.56; −0.71]	−1.88 [−3.14; −0.72]
	HCs	0.08 [−1.13; 1.10)	−1.27 [−3.16; −0.51]	−1.77 [−2.79; −0.37]	−1.91 [−2.92; −0.74]
LFnu-sBP [%]	MS	−0.67 [−9.04; 5.51]	4.75 [5.21; 17.41]	8.47 [2.12; 17.11]	10.09 [2.27; 18.34]
	HCs	−0.45 [−4.56; 3.88]	6.33 [−0.01; 13.26]	13.95 [3.43; 20.50]	17.43 [5.86; 23.81
HFnu-sBP [%]	MS	0.66 [−0.64; 3.71]	1.77 [−1.21; 7.14]	2.35 [0.16; 5.45]	2.29 [0.18; 7.60]
	HCs	1.51 [−0.43; 3.45]	2.69 [0.70; 4.31]	3.17 [0.54; 5.68	4.03 [0.65; 5.75]
PSD-sBP [mmHg^2^]	MS	−1.18 [−5.05; 0.15]	−2.68 [−7.89; 0.07]	−3,27 [−4.90; −1.25]	−3.36 [−5.62; −1.39]
	HCs	−1.63 [−3.69; −0.71]	−3.00 [−5.95; −1.50]	−2.71 [−6.00; −1.88]	−3.21 [−5.85; −1.87]
BRS [ms/mmHg]	MS	−4.93 [−28.72; 3.77]	−4.92 [−24.30; 5.13]	−6.60 [−12.99; −0.44]	−6.39 [−12.10; −0.72]
	HCs	−5.13 [−10.68; −1.30]	−7.12 [−10.81; −2.07]	−8.12 [−11.92; −1.79]	−7.96 [−12.66; −2.60]

HR—heart rate; sBP—systolic blood pressure; dBP—diastolic blood pressure; mBP—mean blood pressure; CI—cardiac index; TPRI—total peripheral index; LFnu-RRI—low-frequency R-R interval in normalized units; HFnu-RRI—high-frequency R-R interval in normalized units; PSD-RRI—power spectral density R-R interval; LF/HF—ratio between low and high band for heart rate and blood pressure variability; PSD-sBP—power spectral density of systolic blood pressure variability; LFnu-sBP—low frequency of systolic blood pressure variability in normalized units; HFnu-sBP—high frequency of systolic blood pressure variability in normalized units; PSD-dBP—power spectral density of diastolic blood pressure variability; LFnu-dBP—low frequency of diastolic blood pressure variability in normalized units; HFnu-dBP—high frequency of diastolic blood pressure variability in normalized units; BRS—baroreflex sensitivity statistically significant differences are indicated with * *p* < 0.05, ** *p* < 0.01, and *** *p* < 0.001.

## Data Availability

The datasets generated during and/or analyzed during the current study are available from the corresponding author on reasonable request.

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
