# Peer review of "Association between Daytime Sleepiness, Fatigue and Autonomic Responses during Head-Up Tilt Test in Multiple Sclerosis Patients"

_brainsci, 2023, doi:10.3390/brainsci13091342_

Round 1
Reviewer 1 Report
The manuscript by Monika Zawadka-Kunikowska et al. compares autonomic responses in patients with MS exhibiting daytime sleepiness and fatigue. They show that MS patients subjected to HUTT present with significantly reduced BP response and reduced cardiac index. They also correlate daytime sleepiness and blood pressure regulation in the patients. Overall, this is an interesting study that opens many directions for future studies.
Authors can address a few concerns to make the manuscript stronger:
1. The introduction needs a better structure. It can benefit from rewriting. In its current state, it lacks coherence. If MS is the subject of research here, it needs to be evident in the story. Right now, the introduction makes it seem that cardiovascular condition is the subject and not MS. Also, try to write in a better way why the said parameters were measured in MS patients.
2. There are too many references in the introduction, and they are not cited properly. For instance, in the introduction paragraph 1, the end of the sentence shows references 4-11. Try to use only the original article next to each condition so that readers don't have to hunt as to which reference talks about OI/dizziness/fatigue/etc
3. Is there any rationale behind using the mentioned time points after HUTT in the study design?
4. Figure 1 is very busy, and the fonts are too small to read. Authors can even divide it into 2 figures.
5. Discussion is too long, and authors try to draw conclusions way beyond the actual results. This a nice study, and a crisp and direct discussion is better than making a stretch.
Author Response
The manuscript by Monika Zawadka-Kunikowska et al. compares autonomic responses in patients with MS exhibiting daytime sleepiness and fatigue. They show that MS patients subjected to HUTT present with significantly reduced BP response and reduced cardiac index. They also correlate daytime sleepiness and blood pressure regulation in the patients. Overall, this is an interesting study that opens many directions for future studies.
Authors can address a few concerns to make the manuscript stronger:
- The introduction needs a better structure. It can benefit from rewriting. In its current state, it lacks coherence. If MS is the subject of research here, it needs to be evident in the story. Right now, the introduction makes it seem that cardiovascular condition is the subject and not MS. Also, try to write in a better way why the said parameters were measured in MS patients.
Answer: We thank the reviewer for that comment, we rewrite Introduction as You suggested.
- There are too many references in the introduction, and they are not cited properly. For instance, in the introduction paragraph 1, the end of the sentence shows references 4-11. Try to use only the original article next to each condition so that readers don't have to hunt as to which reference talks about OI/dizziness/fatigue/etc
Answer: We thank the reviewer for your valuable comment. We have made improvements to the use of citations in the Introduction paragraph as suggested.
- Is there any rationale behind using the mentioned time points after HUTT in the study design?
Answer: We thank the reviewer for your valuable comment. We have added more information about the rationale in this study.
The majority of previous studies have focused on cardiac autonomic parameters only, such as HR and/or BP fluctuations. The use of additional cardiovascular parameters, based on impedance cardiography (ICG), provides a more comprehensive assessment in analyzing the results of the HUTT. Progressive monitoring, through various time points, may indicate abnormal changes in cardiac function or peripheral resistance and highlight compensatory adjustments in other functions [Feng 2018]. Based on these considerations, a greater understanding of the interplay of the sequence of beat-to-beat hemodynamic and autonomic responses during orthostatic challenge and their rela-tionship with fatigue and daily sleepiness symptoms is crucial for enhancing cardio-vascular outcomes. Thus, we aimed to assess dynamic changes in hemodynamic and autonomic function in response to the head-up tilt test in patients with MS, and com-pare it to HCs and evaluate its relationship with patients' reported daily sleepiness and fatigue symptoms.
- Figure 1 is very busy, and the fonts are too small to read. Authors can even divide it into 2 figures.
Answer: We thank the reviewer for the valuable remark. The figure 1 has been changed as you suggested.
- Discussion is too long, and authors try to draw conclusions way beyond the actual results. This a nice study, and a crisp and direct discussion is better than making a stretch.
Answer: We thank the reviewer for your valuable suggestion. We have shortened the Discussion section as you suggested
Reviewer 2 Report
Thank you for the opportunity to review this manuscript. Generally speaking, the manuscript is well written and the methods are very good.
Strenghts: The paper has reviewed exiting evidence and compares with the current evidences. Also the methods section clearly addresses what the study objective and its analysis to lead to the results or anticipated outcomes of the review. The results are well described.
Still there are some statements which require more attention from the authors:
- please rearrange the Introduction and also the reference list (the bilbiographic indices are not in order; also after 24, you have [24, kotsuraki] please modify.
- please rearrange the first two paragraphs from the disscusion part.
Author Response
Reviewers' comments to Authors:
Reviewer: 2
Thank you for the opportunity to review this manuscript. Generally speaking, the manuscript is well written and the methods are very good.
Strenghts: The paper has reviewed exiting evidence and compares with the current evidences. Also the methods section clearly addresses what the study objective and its analysis to lead to the results or anticipated outcomes of the review. The results are well described. Still there are some statements which require more attention from the authors:
- please rearrange the Introduction and also the reference list (the bilbiographic indices are not in order; also after 24, you have [24, kotsuraki] please modify.
Answer: We thank the reviewer for the valuable remark. The introduction has been changed as you suggested.
- please rearrange the first two paragraphs from the disscusion part.
Answer: We thank the reviewer for the valuable comment. The first two paragraphs in the Discussion section have been changed as you suggested.
MS patients displayed lower blood pressure response coupled with a more pronounced decrease in cardiac index, a decreased sympathovagal ratio, and an altered diastolic blood pressure variability in response to orthostatic challenge when compared to HCs.
Reviewer 3 Report
Authors report a study they conducted to evaluate the relationship of multiple sclerosis (MS) patients' reported daytime sleepiness and fatigue with the hemodynamic and autonomic function changes during the head-up tilt test (HUTT). Authors found that orthostatic challenge results in lowered blood pressure response, cardiac index, and an altered blood pressure variability in MS patients. These cardiovascular changes are dependent on disease variant and fatigue levels whereas do exhibit any substantial association with daytime sleepiness. Overall, the findings of current manuscript highlight and validate the previously known clinical significance of daytime sleepiness and fatigue symptoms and related blood pressure regulation in MS patients.
Overall, the current manuscript is presented in a well-structured manner and serves a valuable addition to the earlier reports from the group (Rzepiński et al., Neurol Sci 2022, Zawadka-Kunikowska et al., J. Clin. Med. 2020). The introduction provides a clear overview of the topic, the methods are well-described, and the quantitative analyses are sound. The results are presented in a clear and concise manner, and the discussion is insightful. The authors have cited relevant and recent references throughout the manuscript.
Although I do not have any major revisions to suggest, however, I do have a few minor, yet important concerns. Please provide comment and wherever necessary add to the manuscript as well.
1. It is puzzling that the authors chose to use the 2010 McDonald and Polman et al. criteria to diagnose MS in the current manuscript, even though they used the 2017 revisions of the McDonald criteria in their previous publication (Rzepiński et al., Neurol Sci 2022). This is especially concerning given that the two manuscripts appear to use the same cohort of patients (based on the demographics, including the similar sample size of subgroups of MS variants).
2. Similarly, the authors used different frequency ranges for the LF and HF bands in the current manuscript than they did in their Rzepiński et al., 2022 study. In the current manuscript, the LF band is 0.05-0.15 Hz and the HF band is 0.15-0.40 Hz. In the 2022 study, the LF band was 0.04-0.15 Hz and the HF band were 0.17-0.40 Hz. This is a minor variation, but it is nonetheless concerning.
3. The authors used a 7-minute nd 20-second head-up tilt test (HUTT) in the current manuscript, as opposed to a 5-minute HUTT in their Rzepiński et al.,2022 study. This is a significant change, and the authors have not provided any justification for it.
4. The authors here in the limitations, state that they did not analyze the PPMS and RRMS variants separately due to age differences between the two groups (Age, mean (years) 47.45±11.90). However, in their Rzepiński et al., 2022 study (Age, years 45.8 ± 10.9), they did perform such an analysis and found a relationship between MS variants and fatigue severity with cardiovascular and autonomic parameters. The authors should provide more information about the age difference between the PPMS and RRMS samples in the current manuscript. It is relevant particularly considering the similar demographics between the two current and previous study from the group.
5. In patients with MS, fatigue is more prevalent and severe than daytime sleepiness which makes it easier to overlook. Since authors here report that within the MS patient group, subjective daytime sleepiness exhibited no association with fatigue symptoms, do authors think that including objective assessments of daytime sleepiness for example maintenance of vigilance or sustained attention tasks, pupillography, or wakefulness test, could have made the association significant?
Author Response
Reviewers' comments to Authors:
Reviewer: 3
- Authors report a study they conducted to evaluate the relationship of multiple sclerosis (MS) patients' reported daytime sleepiness and fatigue with the hemodynamic and autonomic function changes during the head-up tilt test (HUTT). Authors found that orthostatic challenge results in lowered blood pressure response, cardiac index, and an altered blood pressure variability in MS patients. These cardiovascular changes are dependent on disease variant and fatigue levels whereas do exhibit any substantial association with daytime sleepiness. Overall, the findings of current manuscript highlight and validate the previously known clinical significance of daytime sleepiness and fatigue symptoms and related blood pressure regulation in MS patients.
- Overall, the current manuscript is presented in a well-structured manner and serves a valuable addition to the earlier reports from the group (Rzepiński et al., Neurol Sci 2022, Zawadka-Kunikowska et al., J. Clin. Med. 2020). The introduction provides a clear overview of the topic, the methods are well-described, and the quantitative analyses are sound. The results are presented in a clear and concise manner, and the discussion is insightful. The authors have cited relevant and recent references throughout the manuscript.
- Although I do not have any major revisions to suggest, however, I do have a few minor, yet important concerns. Please provide comment and wherever necessary add to the manuscript as well.
- It is puzzling that the authors chose to use the 2010 McDonald and Polman et al. criteria to diagnose MS in the current manuscript, even though they used the 2017 revisions of the McDonald criteria in their previous publication (Rzepiński et al., Neurol Sci 2022). This is especially concerning given that the two manuscripts appear to use the same cohort of patients (based on the demographics, including the similar sample size of subgroups of MS variants).
Answer: We thank the reviewer for the valuable remark. It was about the 2017 McDonald criteria, definitely. The Methods section has been improved.
- Similarly, the authors used different frequency ranges for the LF and HF bands in the current manuscript than they did in their Rzepiński et al., 2022 study. In the current manuscript, the LF band is 0.05-0.15 Hz and the HF band is 0.15-0.40 Hz. In the 2022 study, the LF band was 0.04-0.15 Hz and the HF band were 0.17-0.40 Hz. This is a minor variation, but it is nonetheless concerning.
- Answer: We thank the reviewer for the valuable comment. The Methods section has been improved.
- The authors used a 7-minute nd 20-second head-up tilt test (HUTT) in the current manuscript, as opposed to a 5-minute HUTT in their Rzepiński et al.,2022 study. This is a significant change, and the authors have not provided any justification for it.
Answer: We thank the reviewer for the valuable comment. The entire duration of the HUTT protocol was 7 minutes and 20 seconds, while in the study by Rzepiński et al. 2022, results from a 5-minute test were presented. Progressive monitoring of & min of tilt was employed to better understand the physiology, mechanisms, or responses associated with changes in posture and blood pressure regulation. Progressive monitoring, through various time points, may indicate abnormal changes in cardiac function or peripheral resistance and highlight compensatory adjustments in other functions. Based on these considerations, a greater understanding of the interplay of the sequence of beat-to-beat hemodynamic and autonomic responses during orthostatic challenge and their relationship with fatigue and daily sleepiness symptoms is crucial for enhancing cardiovascular outcomes. In the current study, we added this information (Introduction paragraph) as the rationale for gaining a greater understanding of the physiological mechanisms.
- The authors here in the limitations, state that they did not analyze the PPMS and RRMS variants separately due to age differences between the two groups (Age, mean (years) 47.45±11.90). However, in their Rzepiński et al., 2022 study (Age, years 45.8 ± 10.9), they did perform such an analysis and found a relationship between MS variants and fatigue severity with cardiovascular and autonomic parameters. The authors should provide more information about the age difference between the PPMS and RRMS samples in the current manuscript. It is relevant particularly considering the similar demographics between the two current and previous study from the group.
- Answer: We thank the reviewer for the valuable comment. In the study presented, there was no age difference between MS and HCs (p>0.05). The MS group consisted of both PMS and RRMS, with patients in the RRMS group, compared to the PMS group, being significantly younger (40.79±10.78 vs. 52.08±8.09, p<0.001). Therefore, age, along with other predictors (sleepiness measured by ESS, fatigue assessed by CFQ, disease variant (RRMS vs. PMS), and EDSS), was included in the multifactorial analysis. The study aimed to achieve a deeper understanding of the interplay between beat-to-beat hemodynamic and autonomic responses, its sequence during orthostatic challenge and their relationship with fatigue and daily sleepiness symptoms. In the limitations section, it was noted that subgroup analysis between MS variants regarding cardiovascular autonomic responses was not conducted, despite the presence of significant differences in age and disease variant, mainly because this analysis had already been described previously (Rzepiński et al. 2022). We also revised the study limitations to enhance their clarity.
- In patients with MS, fatigue is more prevalent and severe than daytime sleepiness which makes it easier to overlook. Since authors here report that within the MS patient group, subjective daytime sleepiness exhibited no association with fatigue symptoms, do authors think that including objective assessments of daytime sleepiness for example maintenance of vigilance or sustained attention tasks, pupillography, or wakefulness test, could have made the association significant?
- Answer: We thank reviewer for this interesting question. Recent studies have shown conflicting results. Dubessy et al. (2021) compared 44 MS patients, of whom 43.2% experienced fatigue and sleepiness, 34% had only fatigue, and 22.7% had neither fatigue nor sleepiness, with 24 HCs who had fatigue and sleepiness symptoms. They concluded that central hypersomnia has been diagnosed in 53% of patients with MS who experience sleepiness and fatigue, using a combination of 24-hour PSG and Multiple Sleep Latency Test. Kaynak et al. (2006) reported conflicting findings: there was no correlation between FSS and ESS or sleep latency (MSLT), whereas the subjective measurement such Fatigue Impact Scale exhibited such correlations. Frauscher et al. (2005) compared 61 MS patients with 42 HCs and found no correlation was found between pupillary unrest index and subjective measures of EDS. Based on these reports the association of autonomic function and both objective and self-reported measures of sleepiness should be examined in future studies